# Effect of Process Parameters on Microstructure and Properties of Laser Cladding Ni60+30%WC Coating on Q235 Steel

**DOI:** 10.3390/ma16227070

**Published:** 2023-11-07

**Authors:** Shanshan Wang, Wenqing Shi, Cai Cheng, Feilong Liang, Kaiyue Li

**Affiliations:** 1School of Electronics and Information Engineer, Guangdong Ocean University, Zhanjiang 524088, China; 18037269800@163.com (S.W.); chengc1126@163.com (C.C.); likaiyue0512@163.com (K.L.); 2School of Materials Science and Engineering, Guangdong Ocean University, Yangjiang 529500, China; 3Guangdong Provincial Key Laboratory of Intelligent Equipment for South China Sea Marine Ranching, Guangdong Ocean University, Zhanjiang 524088, China; 13542018087@163.com; 4Naval Architecture and Shipping College, Guangdong Ocean University, Zhanjiang 524088, China

**Keywords:** laser cladding, process parameters, microstructure, wear resistance

## Abstract

A Ni60+30%WC composite coating was prepared on the surface of Q235 steel by utilizing a high cooling rate, small thermal deformation of the substrate material, and the good metallurgical bonding characteristics of laser cladding technology. This paper focuses on the study of the composite coatings prepared under different process parameters in order to select the optimal process parameters and provide theoretical guidance for future practical applications. The macroscopic morphology and microstructure of t he composite coatings were investigated with the help of an optical microscope (OM) and a scanning electron microscope (SEM). The elemental distribution of the composite coatings was examined using an X-ray diffractometer. The microhardness and wear resistance of the composite coatings were tested using a microhardness tester, a friction tester, and a three-dimensional (3D) profilometer. The results of all the samples showed that the Ni60+30%WC composite coatings prepared at a laser power of 1600 W and a scanning speed of 10 mm/s were well formed, with a dense microstructure, and the microhardness is more than four times higher than the base material, the wear amount is less than 50% of the base material, and the wear resistance has been significantly improved. Therefore, the experimental results for the laser power of 1600 W and scanning speed of 10 mm/s are the optimal process parameters for the preparation of Ni60+30%WC.

## 1. Introduction

As an ordinary carbon structural steel, Q235 steel has the advantages of moderate carbon content, good comprehensive properties, and low cost, so it is used extremely widely for parts in industrial production at present [1]. However, it has a low yield point and poor wear resistance, failing extremely easily in special environments, so it is especially important to carry out reinforcement treatment of its surface. At present, the surface modification technology for Q235 steel mainly uses laser cladding technology.

Laser cladding technology is characterized by quick heating and cooling, small thermal deformation of substrate material, low dilution rate, a good metallurgical bond formed between cladding and substrate materials, etc. It also has advantages such as short processing time, flexible operation, and high accuracy [2,3], so it was used to prepare the high performance coating on the surface of Q235. At present, the commonly used coating powders for laser cladding include cobalt-based alloys [4], iron-based alloys [5], and nickel-based alloys [6]. Nickel-based self-fluxing alloy powders are widely used in various industries owing to their low cost and good wear resistance and corrosion resistance at high temperature. As one of nickel-based self-fluxing alloy powders, Ni60 alloy powder has relatively good hardness and deoxidation slaggability and good wettability, and it contains elements B, C, and Cr, which are involved in the synthesis of hard phase, so the Ni60 alloy powder can effectively improve the corrosion resistance of the coating at high temperatures [7,8,9]. Therefore, Ni60A is widely used in the field of surface modification; however, pure nickel alloy has low hardness, which makes it difficult to meet the strength property requirement. To further improve the hardness and wear resistance of nickel-based alloy coatings, some researchers have used the method of adding ceramic powder (WC), rare earth oxide, or reinforcement phase to reduce the friction effect and increase the useful life of the material. Tungsten carbide has relatively high hardness and wear resistance, which, together with the good wettability of nickel-based alloy, are research hot spots for reinforcing nickel-based composite coating [10]. In Reference [11], Ni/WC composite coating, as a laser cladding material, is used on Cr12MoV. Results show that the Ni60+30%WC coating has the best properties, and its wear resistance is improved by ~30% compared with coating without WC added. The hardness of the cladding layer is 67–68 HRC, 30% higher than that of the substrate.

The variable factors in a laser cladding process include the substrate material, cladding material, and process parameters. These three variable factors have different functions in the cladding process, and the variation of each factor will influence the cladding results (e.g., the morphology, properties, etc. of the cladding layer) [12]. The variations in the laser process parameters have relatively large influences on the cladding results, and suitable laser process parameters can improve the properties of the cladding layer to some extent. The main laser process parameters influencing the properties of cladding layer include laser power, scanning speed, laser spot diameter, overlapping ratio, and powder feed quantity [13]. The laser power and scanning speed are two factors having the most significant influences on the shaping quality and geometric characteristics of laser cladding layers [14,15,16]. Tan et al. [17] studied the influence of the scanning speed on the laser cladding Ni60 composite coating on the Ti6Al4V alloy surface and found that increasing the scanning speed can effectively improve the hardness and wear resistance of the cladding layer, but a too high scanning speed may cause defects such as cracks and pores. Kobryn et al. [18] found through study on the influences of laser power on the microstructure, shaping quality, and porosity of the cladding layer that the increase in laser power is favorable for improving the cladding layer against its crack and pore defects. Sun et al. [19] studied the influence of laser energy density on Ni/WC composite coating, providing data reference for the Cr12MoV repair process.

Comprehensive past research results show that the process parameters during laser cladding have a crucial influence on the performance of prepared Ni60 coatings or Ni60/WC composite coatings. However, there are few reports on the properties of Ni60/WC composite coatings prepared by controlling both laser power and laser scanning speed. It is discerned through early experiments [11,20] that the cladding powder Ni60+30%WC can bring about optimal properties. After the cladding powder is determined as Ni60+30%WC, the Ni60+30%WC composite coating was prepared on the surface of Q235 steel by changing the laser power and laser scanning speed, and its microstructure and mechanical properties were studied. In order to investigate the effects of laser power and laser scanning speed on the forming quality, size, organization, and mechanical properties of the composite coatings, we analyze the evolution of the organization and illustrate the intrinsic correlation between different organizational features and mechanical properties to find out the optimal combination of laser power and laser scanning speed within the reasonable level of each index.

## 2. Materials and Methods

### 2.1. Experimental Materials and Equipment

The experimental substrate material was Q235 carbon tool steel, with chemical composition (mass fraction) shown in Table 1, and its dimensions were 90 × 50 × 3 (L × W × H, in mm). Before laser cladding, the substrate was ground with sandpapers with different grit sizes in turn to remove the oxide film on the surface, and then rinsed with absolute alcohol; afterwards, it was dried for subsequent use.

The mixed powder of spherical WC (with a purity of 99.8%) and Ni60 (with a mean particle size of 15–53 μm) was chosen as the cladding material, and based on the preliminary test, WC powder and Ni60 powder were mixed according to the mass fraction ratio of 3:7 [10]. The chemical composition (mass fraction), oxygen and nitrogen contents, and particle size distribution of the Ni60 powder are shown in Table 2, Table 3 and Table 4. Figure 1 shows the micromorphology of Ni60 and WC powders.

In this laser cladding experiment, the experimental equipment selected was an XL-F2000W optical fiber laser cladding system, of which the laser power threshold was 0–2000 W, defocusing was +5 mm, the powder feed method was a pre-coating method, and the preset thickness was 1 mm. It was selected to change the laser power P and scanning speed V out of the process parameters, to design experiments (Table 5). Single-pass and multi-pass cladding experiments were carried out one after another, to explore the influences of process parameters on shaping dimensions, macromorphology, microstructure, microhardness, and friction wear property, finally obtaining the optimal process parameter values for good shaping and excellent properties.

### 2.2. Structure Characterization and Mechanical Properties Test

The sample obtained by laser cladding was cut to take specimens. The sample was wire cut along the direction perpendicular to the laser scanning direction, yielding a cross-section specimen of cladding layer, which was ground, polished and chemically corroded to form a metallographic specimen. The metallographic specimen was corroded by metallographic etchant.

The width (w), height (h), and depth (d) of the cladding layer were measured with a VHX-700 series Keyence digital microscope (Keynes (China) Co., Ltd., Shanghai, China). The microstructure was observed with a Tescan Mira4 field emission SEM. Phase analysis of the specimen was conducted with a Shimazu XED-6100 X-ray diffractometer (Beijing, China) at the following specific parameters: Cu-Kα radiation was used, the tube current and tube voltage were 40 mA and 40 kV, respectively, the scanning speed was 6°/min, and 10–90° coupled continuous scanning was used with a step of 0.02°. The microhardness of the cladding layer was tested with a digital microhardness tester (MHVD-1000AT, Shanghai Jujing Precision Instrument Manufacturing Co., Ltd., Shanghai, China) produced by Shanghai Jvjing Precision Instrument Manufacturing Co., Ltd. The microhardness indenter type was Vickers. The microhardness was tested once per 0.2 mm in the layer depth direction, the test load was 200 gf, the loading time was 10 s, and the microhardness results at each point were measured three times. Reciprocating wear test was conducted on a CETR-UMT-2MT friction testing machine (American CETR Friction and wear Testing Machine Co., Ltd., San Jose, CA, USA), and the test parameters were as follows: the grinding balls were made of silicon nitride (Si_3_N_4_) and had a diameter of 3 mm, the test load was 10 N, the frequency was 41 mm/s, and the test time at room temperature was 1800 s. The wear quantity of the specimen was measured before and after wear, and the measurement accuracy was 0.01 mg. The mass difference before and after wear was measured for each group of specimens.

## 3. Results and Discussion

### 3.1. Single-Pass Cladding Macromorphology

Macromorphology is an important basis for determining whether process parameters are rational and plays a great role in testing the quality of the cladding layer. Figure 2 shows the macromorphology of nine groups of specimens obtained through single-pass cladding experiments. It can be discerned that all specimens form close bonding with the corresponding substrate materials, with surfaces having good overall flatness.

### 3.2. Macromorphology and Shaping Dimensions of Single-Pass Cladding Cross-Section

The quality of single-pass cladding macromorphology is influenced by the process parameters (laser power P and scanning speed V), and the melting width w, melting height h, width–height ratio w/h, and dilution rate η are important indicators measuring the single-pass cladding quality. The dilution rate is related to the bonding strength between cladding layer and substrate material. In a laser cladding process, the laser energy required for cladding the coating powder onto the substrate material is fixed. When there is surplus laser energy generated, it will be absorbed by the substrate material. After the substrate material absorbs the energy, its surface will be dissolved, thus diluting the components of the cladding coating. The dilution rate is used to characterize the degree of dilution of the cladding coating material by the substrate material [23]. If the dilution rate is too high, it will reduce the cladding coating surface reinforcement effect. If the dilution rate is too low, it will lower the bonding strength between coating and substrate, and it is even easy to cause the coating to peel off [24]. Dilution rate can be calculated using a geometric method, in other words, the dilution rate η can be calculated based on the measurement values of melting height and melting depth, with the calculation formula as shown in Equation (1). Figure 3 shows the morphology of a cross-section of the cladding specimen obtained through single-pass cladding experiments, and the experiment scheme and results are shown in Table 6. Figure 4a–c are obtained from the experimental results in Table 6, and they are used to observe the separate influence mechanisms of the process parameters (laser power P and scanning speed V) on the melting width w, melting height h, and width–height ratio w/h of the cladding layer, as well as the relationships of the process parameters with the dilution rate.
(1)$$\eta =\frac{d}{h+d}\times 100\%$$

(1) Through Figure 4a,b, together with Table 6, it can be discerned that when the laser power increases from 1600 W to 2000 W, the melting width and melting height increase accordingly; however, when the laser power is 1200 W, the above law is not satisfied. The reason may be that a too low laser power causes the heat inputted by the laser beam not to reach the degree of full melting of the cladding powder, so the metal powder is not melted well. In the case of a constant scanning speed, the influences of the change in laser power on the cladding layer dimensions and dilution rate are not very evident.

(2) As can be seen from Figure 3, there are relatively evident particles in specimens 3, 6, and 9 at the scanning speed of 15 mm/s. The reason may be that a too high scanning speed causes the cladding powder not to well absorb the laser energy for melting, so some unmelting powder particles form a solid solution with the cladding layer. It can be seen from Figure 4a,b that, with the increase in scanning speed, the melting width exhibits a trend of decreasing first and then increasing as a whole, with the increasing trend being relatively gentle. The melting height exhibits a trend of increasing first and then decreasing as a whole, with the decreasing trend being relatively gentle. In the case of constant laser power, the scanning speed determines the laser beam irradiation time for the cladding powder and substrate material. The change in scanning speed may cause the change of powder melting per unit area and may also influence the absorption of energy by substrate material. When the scanning speed is relatively low, there may be HAZs formed. For example, in the macromorphology shown in Figure 2, specimen 7 exhibits a relatively evident HAZ, which further changes the dilution rate. Therefore, the dilution rate can be regulated in a suitable range by adjusting the scanning speed.

(3) As can be discerned from Figure 4b,c, the width–height ratio is inversely proportional to the melting height. In a multi-pass cladding experiment, defects easily occur in the overlapped bottom, and increasing the width–height ratio can improve the wettability of the cladding layer, thus effectively reducing such defects [25]. However, a too large width–height ratio may cause an increase in the dilution rate. For example, specimens 1, 4, and 7 at identical scanning speed 5 mm/s have an evidently higher width–height ratio and melting depth than the other two specimens at the same laser power, with melting depth reaching 0.58 mm maximum and the difference in melting depth reaching 0.49 mm, causing these three specimens to have a too high dilution rate. A high dilution rate may result in poor metallurgical bond of the cladding layer with the substrate material, reducing the reinforcement property of the coating.

In summary, both process parameters, i.e., scanning speed V and laser power P, can change the cladding layer dimensions and dilution rate. In comparison, scanning speed can more significantly change the width–height ratio of the cladding layer and more rationally regulate the dilution rate than laser power, thus having more significant influence on the shaping quality of the cladding layer.

### 3.3. Multi-Pass Cladding Macromorphology

In the laser cladding shaping process, single-pass cladding shaping dimensions may have important influences on the macromorphology of the cladding layer shaped through multi-pass cladding [25]. To better obtain rational process parameter values, thus getting a better cladding layer, multi-pass cladding experiments were carried out for verification according to the experimental scheme in the experiment table. The schematic diagram of scanning paths is shown in Figure 5. In the actual experiments, 12 passes were prepared for each group at a pass spacing of 1 mm, and then the surface macromorphology of nine groups of multi-pass experimental results was obtained, as shown in Figure 6.

As can be discerned from Figure 6, the specimens exhibit varying degrees of cracks in the surface whether at one power or at one scanning speed. Specimens a3, a6, and a9 have relatively many residues on their surfaces, with cladding quality lower than other specimens. These three specimens underwent identical scanning speed (15 mm/s). The reason may be that the scanning speed is too high, causing the powder on the substrate surface not to fully melt. Specimen a7 has depressions in its surface, which, together with the single-pass cladding macromorphology of specimen 7 shown in Figure 1 and the melting depth value shown in Table, demonstrate that when the laser power is relatively high and the scanning speed is relatively low, the energy inputted into the molten pool by the laser beam is relatively high, the melting depth is relatively large, and the convection and evaporation effects of the metal melting in the molten pool are enhanced, thus causing defects occurring in multi-pass cladding overlapping.

### 3.4. Laser Energy Density

In a laser cladding shaping process, the shaping macromorphology of the cladding layer is jointly influenced by multiple process parameters. The laser energy input varies with the combination of parameter values. The macromorphology of the cladding layer is closely related to the energy density inputted into the cladding layer by the laser in the cladding process. Changing the laser power and scanning speed can directly influence the energy density, thus affecting the properties of the cladding layer. To measure the quantity of energy inputted into the cladding layer by the laser beam at different combinations of laser power and scanning speed with other conditions unchanged, a relational expression of the laser power, scanning speed, and pass spacing with the energy density inputted by the laser beam into per unit area of cladding layer was established, as shown in Equation (2):(2)E=P/(V×H)
where E is the laser energy density received per unit area of cladding layer, P is the laser power, V is the scanning speed, and H is the pass spacing. It can be discerned from the equation that the laser energy density received by the cladding layer increases with the increase in laser power but decreases with the increase in scanning speed [25].

The energy density E corresponding to each group of parameters in the experiments was calculated, and the results are shown in Table 7. Specimen a3 has the lowest energy density: 80.0 J, specimen a7 has the highest energy density: 400.0 J, and specimen a5 has an intermediate energy density value: 160.0 J.

### 3.5. Microstructure Analysis

Different laser energy densities generate different temperatures of molten pool, further influencing the microstructure morphology of the cladding layer. Based on the laser energy density values calculated in multi-pass cladding experiments, the single-pass cladding specimens with the highest, intermediate, and lowest energy density values, which were S3, S5, S6, S7, respectively, were selected for microstructure observation. The full view of the upper, middle, and lower parts of the cladding layer were photographed with a SEM for these four specimens.

Shown in Figure 7(a1–d1) are the full views of the coatings of S3, S5, S6, S7, respectively. It can be seen that there is a fusion line formed between the coating and substrate in each specimen, indicating that there has been good metallurgical microstructure formed between the coating and substrate; however, there are relatively many particles seen in Figure 7(a1(S3),(c1(S6)), these particles are due to the following decomposition reaction of some WC in the molten pool under the action of laser heat. Thermal decomposition is shown in Equations (3) and (4) [26]:(3)$$2WC=W_{2}C+C$$
(4)$$W_{2}C=2W+C$$

The W and C elements produced by WC decomposition will be solidly dissolved in the matrix and play the role of solid solution strengthening. At the same time, during the cooling process, W and C elements and Cr, Si, and B elements separated from Ni60 powder form a new phase and then precipitate again, dispersing in the matrix and playing the role of precipitation strengthening [27]. When enlarging the view in Figure 8a3 (S3 laser cladding middle)and Figure 8c2 (S6 laser cladding top), it is found that there are massive and fishbone secondary carbides in the edge and surrounding area of WC. These secondary carbides are the secondary precipitated phase produced by WC decomposition, usually belonging to the brittle phase.

Observation of Figure 8d4 (S7 laser cladding bottom) shows where columnar dendrites are formed at the bonding interface near the bottom of the coating. According to the solidification theory, the morphology of the solidification structure is determined by the crystallization parameter G/R (G is the temperature gradient, R is the solidification rate) [28,29]. The growth direction of columnar dendrites controlled by the temperature gradient is exactly opposite to the direction of heat flow. Theoretically, the direction of heat flow and the binding interface are perpendicular to each other, so the columnar dendrites continue to grow in the middle or upper part of the coating along the direction perpendicular to the binding interface [30]. There is a crack defect in Figure 8d2 (S7 laser cladding top); this is because the laser power is large, the scanning speed is slow, and the laser energy density is high. At this time, the temperature of the liquid molten pool will increase sharply, resulting in the expansion of the temperature difference between the cladding layer and the matrix, which easily causes local thermal stress concentration and increases the crack sensitivity. The cladding layer prepared by high laser energy density easily produces cracks and other defects, which will affect the coating quality.

Since both too high and too low laser energy density can cause the microstructure to exhibit a coarsening trend, only a proper laser energy density selected can form microstructure with good metallurgical quality and uniform refinement, further yielding optimal mechanical properties [31]. In comparison, S5 has better shaping quality and evener microstructure distribution.

According to the above analyses of shaping dimensions, dilution rate, laser energy density, and microstructure, etc. based on the single-pass and multi-pass cladding experiment results, it is temporarily believed that when the process parameters are selected as 1600 W of laser power and 10 mm/s of scanning speed, the above properties are better.

### 3.6. Phase Analysis of Coating

Similarly, single-pass cladding S3, S5, S6, and S7 were selected to analyze the phase composition of the composite coating, and the XRD patterns of composite coating shown in Figure 9 were obtained for analysis. It can be seen that the peaks differ among the coatings. The reason is that the laser energy density varies with process parameter values [32], and there are fully melting powder and non-fully melting powder phenomena occurring in the cladding layer at different process parameter values, causing different element contents in different coatings. A high-energy laser beam can make the cladding powder and the substrate surface melt simultaneously [33]. As can be discerned from the figure, the phases of the cladding layers mainly include γ-Ni, Ni3Fe, CrSi2, Ni2Si, CrB2, and W2C. The non-detection of the physical phase of WC may be due to the fact that the detection surface is the surface of the sample, since the density of WC is larger than that of Ni60, most of the WC particles sink to the bottom of the molten pool [34], and a small number of WC particles remaining on the surface, as shown in Equations (3) and (4), are decomposed by heat, so that the peak of WC is not detected on the surface. The positions of the diffraction peaks roughly do not vary with the specimen, but their intensities vary with the specimen. In particular, for S3 and S6, the diffraction peaks of CrB2 and W2C at around 35° and the diffraction peak of Ni2Si at around 46° have relatively high intensities, combined with SEM in Figure 8a3,c2, it is due to the enrichment of secondary precipitates at the grain boundaries.

### 3.7. Microhardness Analysis

Figure 10a shows the changes in microhardness of nine groups of single-pass cladding specimens from cladding layer to Q235 steel substrate in experiments. Figure 10b shows the mean value of the sample cladding layer. It can be seen that the hardness of the cladding layer is higher than that of substrate in all of nine groups of specimens. S1, S4, and S7, as different powers at the same scanning speed, have large differences in hardness values and lower hardness compared with other samples. The main reason is that when the scanning speed is slow, the WC particles are more likely to decompose, and the corresponding WC hard phase is reduced, resulting in a lower increase in coating hardness.

S6 has the highest hardness. However, according to Figure 7c1, there are large number of secondary carbides due to the thermal decomposition of WC particles. A combination of the SEM images in Figure 8a3,c2 and the XRD pattern in Figure 9 shows the places with higher peaks in S3 and S6, indicating that there are more secondary carbides, so the high hardness may be caused by the accumulation of a large number of secondary carbides. The main strengthening methods include solid solution strengthening and precipitation strengthening.

### 3.8. Wear Test Analysis

Figure 11 shows the curves of the friction coefficient of the coatings and substrate. Curve S0 represents the friction coefficient of the substrate. The friction coefficients are 0.508 (S0), 0.499 (S3), 0.387 (S5), 0.414 (S6), and 0.516 (S7), respectively.

As can be discerned from the figure, the friction state varies with the process parameter values; however, the friction undergoes the initial running-in stage and stable wear stage for all specimens. In the initial running-in stage, the contact between grinding pairs is micro-convex body contact [35], with a small contact area and high stress, and the micro-convex body was squeezed to generate a large quantity of ground particles, which have a “plowing” effect on the contact surfaces, so the friction coefficient increases sharply for all specimens [36]. Afterwards, the friction curves of specimens S0, S5, S6, and S7 gradually tend to be balanced, entering a stable wear stage; however, the friction state of specimen S3 is not stable, not reaching a balanced state. Specimen S5 has the smallest friction coefficient, followed by S6. Specimens S5 and S6 were prepared at the same laser power of 1600 W.

Figure 12 shows the wear amount of the sample, which are S0 (0.0269), S3 (0.01145), S5 (0.00605), S6 (0.0086), and S7 (0.0116), respectively. The wear amount of the matrix S0 is the most, and the wear amount of S5 is the least, and the wear amount of S0 is about 4.45 times of that of S5. The wear of coating samples is less than that of substrate, and the wear of S3, S5, S6, and S7 is less than 50% of that of S0. S3 and S6 wear the most in addition to the substrate, combined with the SEM diagram of Figure 8a3,c2, indicating that the decomposition and decarbonization of WC is not conducive to the wear resistance of the composite coating [26].

Wear 3D morphology is also an indicator to measure the wear resistance of a material. S5, which has the best friction coefficient and the least wear amount, was selected to compare the three-dimensional wear morphology with the matrix S0. Figure 13 are diagrams of 3D morphology at the wear marks of specimens S0 and S5. It can be seen from the figures that varying degrees of wear occurs in the surfaces of both specimens. The wear mark of S0 is wider and deeper. Since there is no cladding layer for protection, the substrate is easier to be destroyed, and the surface is easier to come off during wear. The wear mark of S5 is narrower, and the corresponding friction coefficient curve is stabler, it shows that Ni60 powder in the cladding coating has good wettability under the process parameters of S5, and the coating effect is better when combined with WC powder with higher hardness.

Through comprehensive analysis of the wear resistance of specimens based on the friction coefficient, wear quantity, and wear 3D morphology, it is obtained that specimen S5 has the smallest and most stable friction coefficient, relatively low wear quantity, and relatively good 3D morphology, so when the laser power is 1600 W and the scanning speed is 10 mm/s, the prepared specimen coating has better wear resistance.

## 4. Conclusions

In this paper, the influences of process parameters on the laser cladding of Ni60+30%WC composite powder onto Q235 substrate were explored, with the following conclusions drawn:(1)Different process parameter values have different influences on the morphology of the cladding layer. Both laser power P and scanning speed V, out of the process parameters, can change the dimensions and dilution rate of cladding layer. In the experiments this time, the scanning speed can more significantly change the width–height ration of cladding layer and more rationally regulate the dilution rate than the laser power and has more significant influence on the shaping quality than the laser power.(2)Microstructure analysis shows that process parameter values influence the laser energy density, and both too high and too low laser energy density may cause the microstructure to exhibit a coarsening trend. Only rational process parameter values and laser energy density can yield a cladding coating with a dense and refined microstructure and even element distribution.(3)Analysis of coating properties shows that the composite coating exhibits evident improvement in respect to such properties as hardness, wear degree, and wear quantity, compared with the substrate. Through comprehensive analysis of coating properties, it is obtained that the optimal process parameters values are as follows: a laser power of 1600 W and scanning speed of 10 mm/s, and the specimen coating prepared at these process parameter values has better wear resistance.

## Figures and Tables

**Figure 1 materials-16-07070-f001:**
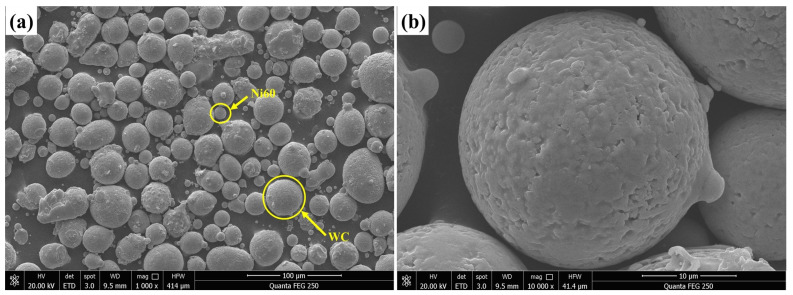
The microstructure of alloy powder: (**a**) Ni60+WC mixed powder; (**b**) Ball WC.

**Figure 2 materials-16-07070-f002:**
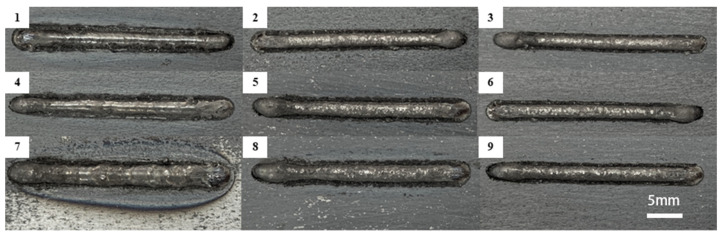
(**1**–**9**) are the macroscopic morphologies of 9 groups of samples prepared by S1–S9 single-pass cladding experiment.

**Figure 3 materials-16-07070-f003:**
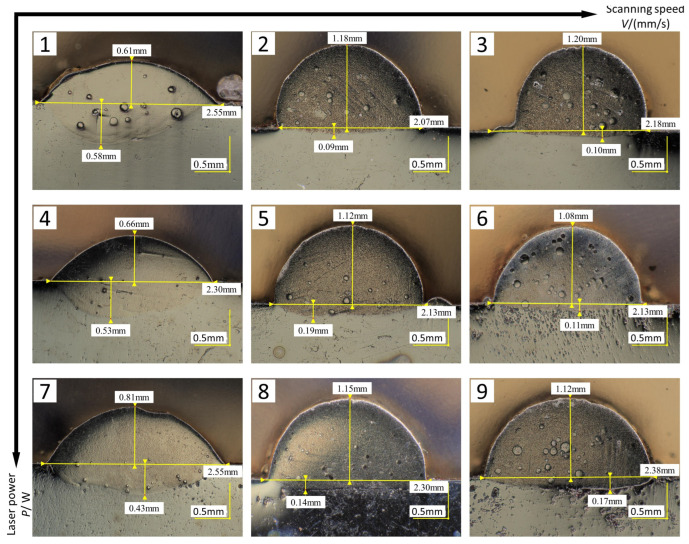
(**1**–**9**) are the cross-section morphologies prepared by single-pass S1–S9 coating experiment.

**Figure 4 materials-16-07070-f004:**
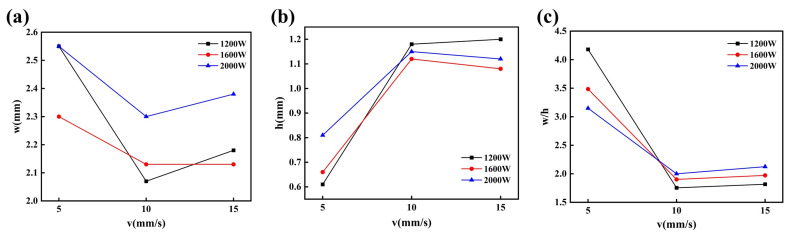
Influence of process parameters on melt width, melting height and aspect ratio: (**a**) effect on melting width; (**b**) effect on melting height; (**c**) effect on melting aspect ratio.

**Figure 5 materials-16-07070-f005:**
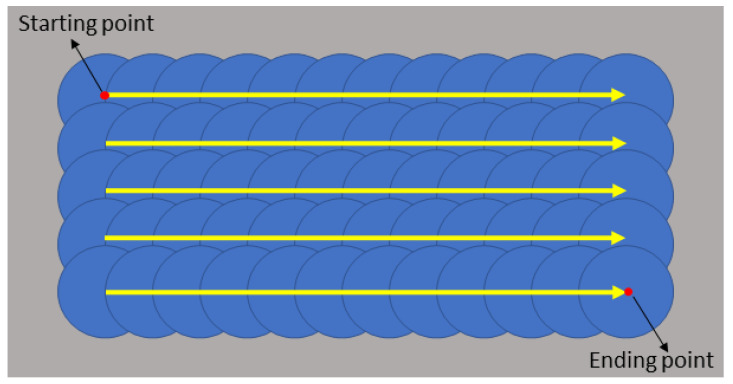
Multi-pass cladding scanning path schematic diagram.

**Figure 6 materials-16-07070-f006:**
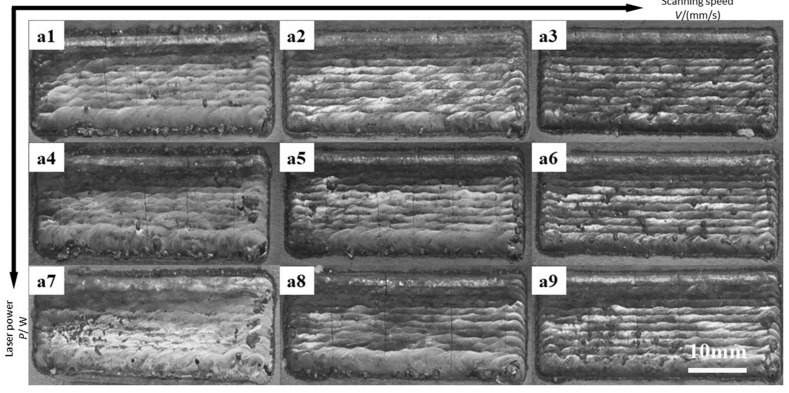
(**a1**–**a9**) are the macroscopic morphologies of 9 groups prepared by S1–S9 through multiple cladding experiments.

**Figure 7 materials-16-07070-f007:**
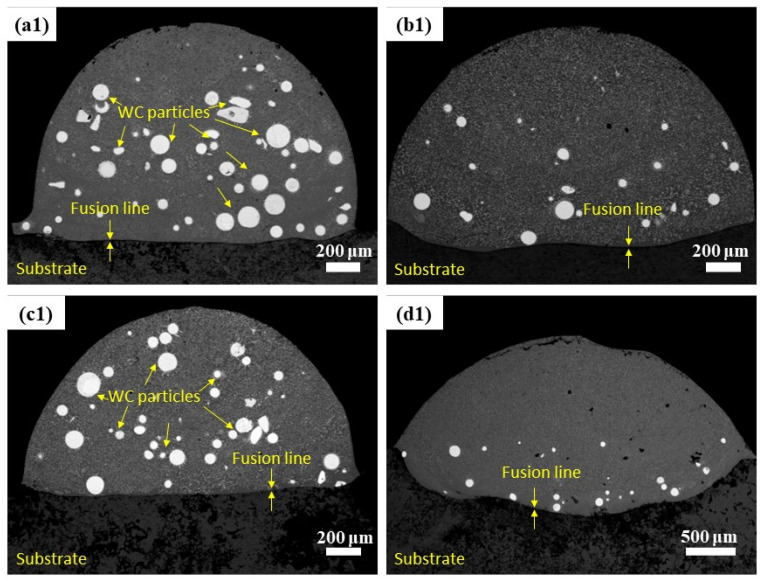
Microstructure of different specimens: (**a1**) S3; (**b1**) S5; (**c1**) S6; (**d1**) S7.

**Figure 8 materials-16-07070-f008:**
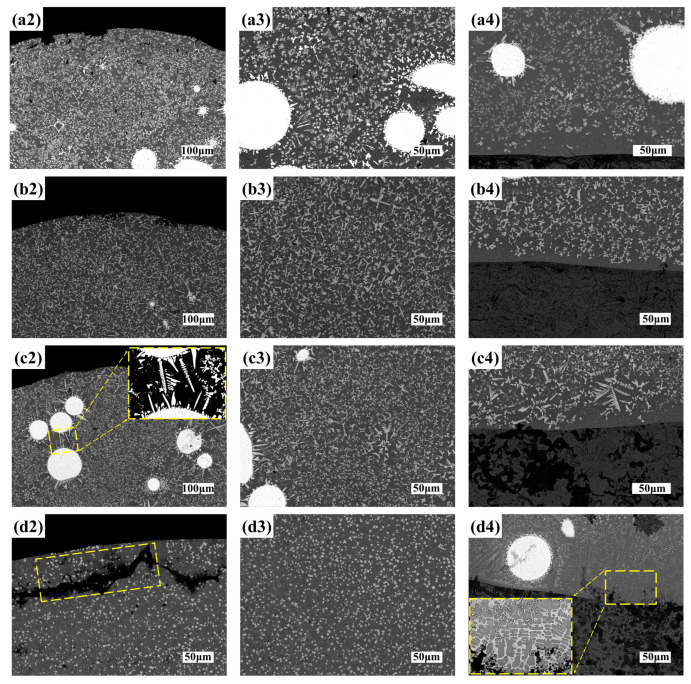
Typical microstructure morphology at different positions of the cladding layer: (**a2**) S3 laser cladding top; (**a3**) S3 laser cladding middle; (**a4**) S3 laser cladding bottom; (**b2**) S5 laser cladding top; (**b3**) S5 laser cladding middle; (**b4**) S5 laser cladding bottom; (**c2**) S6 laser cladding top; (**c3**) S6 laser cladding middle; (**c4**) S6 laser cladding bottom; (**d2**) S7 laser cladding top; (**d3**) S7 laser cladding middle; (**d4**) S7 laser cladding bottom.

**Figure 9 materials-16-07070-f009:**
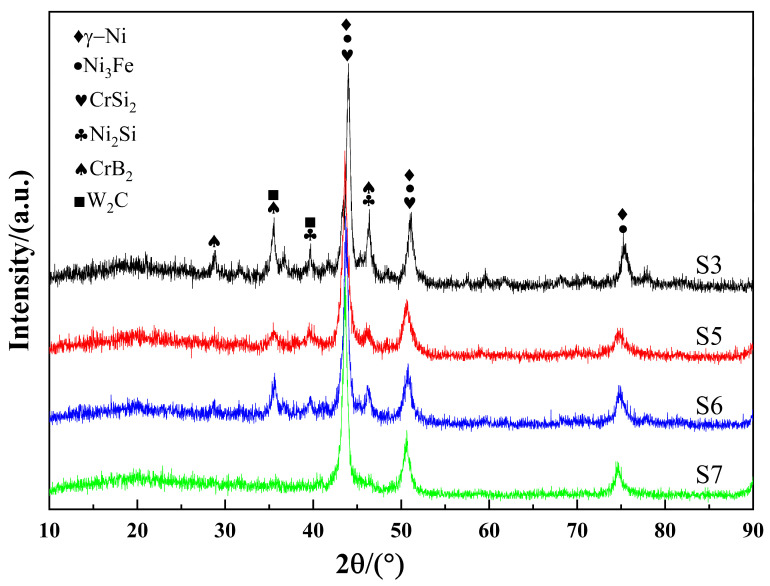
XRD chromatogram of different samples of multi-pass composite coatings.

**Figure 10 materials-16-07070-f010:**
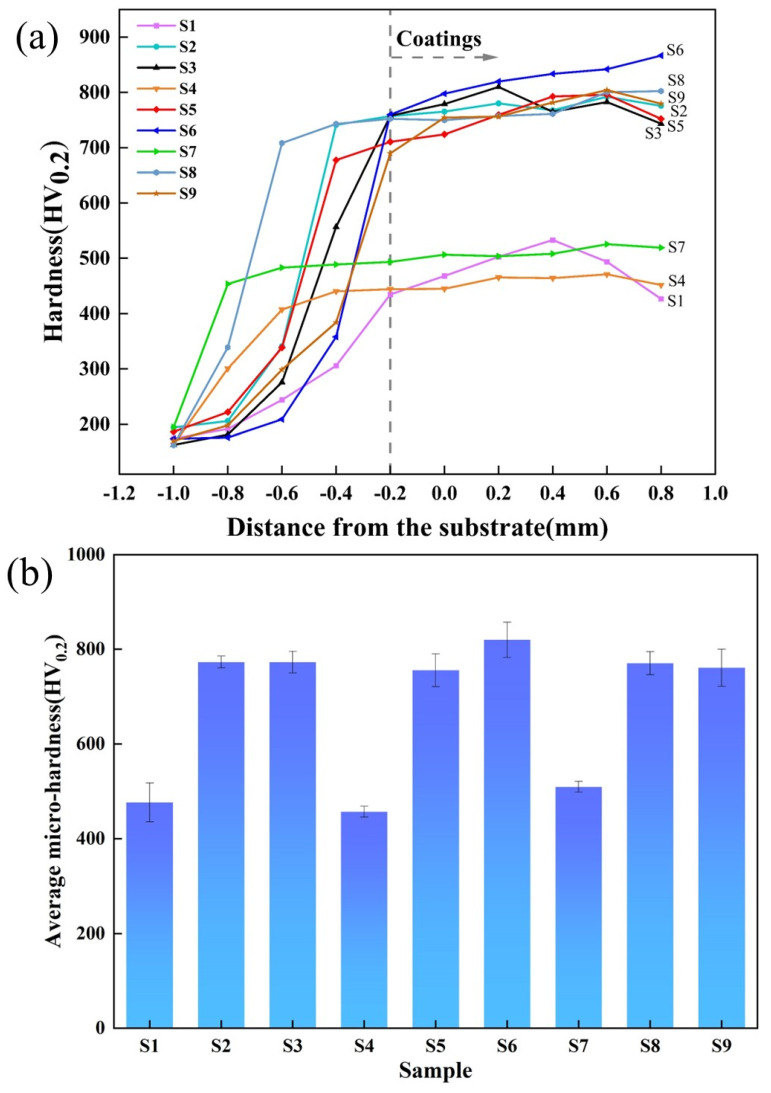
Microhardness change of single-pass cladding sample: (**a**) hardness change from substrate to cladding layer; (**b**) average hardness of cladding layer.

**Figure 11 materials-16-07070-f011:**
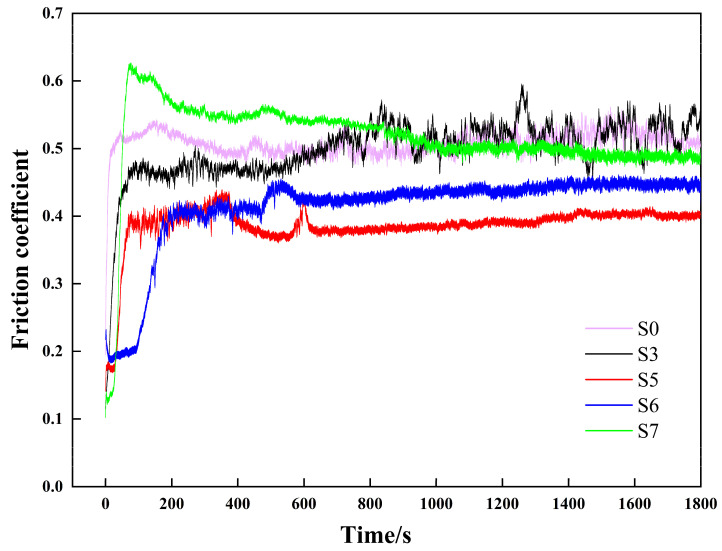
Friction coefficient curve of multi-pass composite coatings.

**Figure 12 materials-16-07070-f012:**
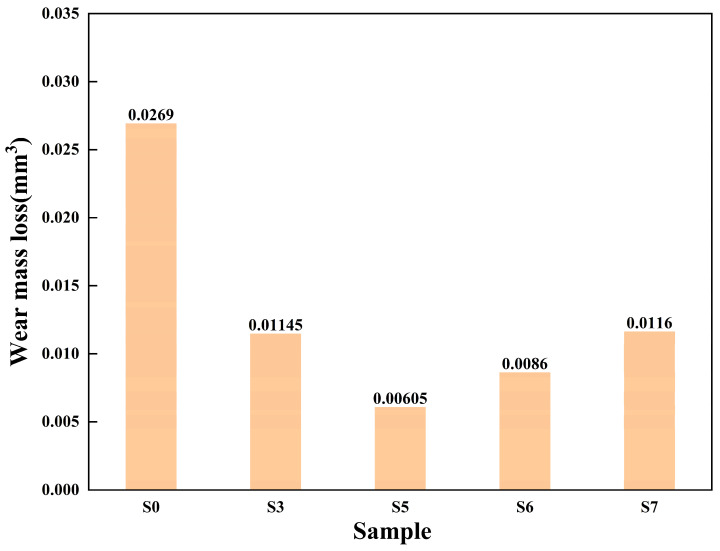
Wear mass loss of multi-pass composite coatings.

**Figure 13 materials-16-07070-f013:**
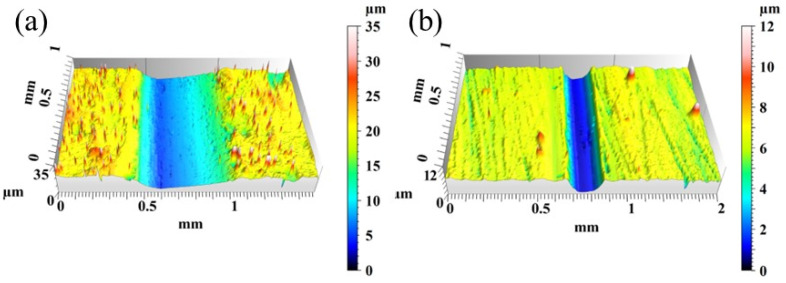
The three-dimensional morphology of the worn surface of the sample: (**a**) S0; (**b**) S5.

**Table 1 materials-16-07070-t001:** Chemical composition of Q235 carbon steel (wt.%).

Type	C	Si	Mn	P	S	Fe
Q235	0.15	0.15	0.25	0.12	0.11	Bal.

**Table 2 materials-16-07070-t002:** Chemical composition of Ni60 powder (wt.%).

Type	Mo	Fe	Cu	Cr	Si	C	B	P	Ni
Ni60	2.93	2.99	3.01	16.01	4.94	0.66	3.90	<0.005	Bal.

**Table 3 materials-16-07070-t003:** Oxygen and nitrogen content of Ni60 powder.

O/ppm	O/ppm	Test Reference Standard
122	17	GB/T 14265 [21]

**Table 4 materials-16-07070-t004:** Particle size distribution of Ni60 powder.

D10/µm	D50/µm	D90/µm	Test Reference Standard
22.1	35.4	56.0	GB/T 19077 [22]

**Table 5 materials-16-07070-t005:** Factor level design table.

Level	Laser Power P/W	Scanning Speed V/(mm/s)	Pass Spacing H/mm
1	1400	5	1
2	1600	10	1
3	2000	15	1

**Table 6 materials-16-07070-t006:** Test plan and experimental results.

Procedure	Laser Power P/W	Scanning Speed V/(mm/s)	Width w/(mm)	Height h/(mm)	Depth d/(mm)	Aspect Ratio w/h	Dilution Rate η/(%)
1	1200	5	2.55	0.61	0.58	4.18	48.74
2	1200	10	2.07	1.18	0.09	1.75	7.09
3	1200	15	2.18	1.20	0.10	1.82	7.69
4	1600	5	2.30	0.66	0.53	3.48	44.54
5	1600	10	2.13	1.12	0.19	1.90	14.50
6	1600	15	2.13	1.08	0.11	1.97	9.24
7	2000	5	2.55	0.81	0.43	3.15	34.68
8	2000	10	2.30	1.15	0.14	2.00	10.85
9	2000	15	2.38	1.12	0.17	2.13	13.18

**Table 7 materials-16-07070-t007:** Test parameters and energy density table.

Procedure	Laser Power P/W	Scanning Speed V/(mm/s)	Pass Spacing H/mm	Energy E/(J/mm^2^)
1	1200	5	1	240.0
2	1200	10	1	120.0
3	1200	15	1	80.0
4	1600	5	1	320.0
5	1600	10	1	160.0
6	1600	15	1	106.7
7	2000	5	1	400.0
8	2000	10	1	200.0
9	2000	15	1	133.3

## Data Availability

The data presented in this study are available on request from the corresponding author.

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
