# Peer review of "Effect of Process Parameters on Microstructure and Properties of Laser Cladding Ni60+30%WC Coating on Q235 Steel"

_materials, 2023, doi:10.3390/ma16227070_

Round 1

Reviewer 1 Report

Comments and Suggestions for Authors

The paper aims to research the important topic, but the investigation itself is relatively weak. I’ve listed some remarks to this paper:

1. [Major] Line 147 „However, specimen 7 has a relatively evident heat-affected zone (HAZ).” – any description of HAZ should be supported by microhardness measurements, how is it possible to conclude on HAZ by looking at top surface of a treated element?

2. [Minor] What was the size of the WC particles? “WC ball” in Fig. 1. has about 20 microns of diameter, but in Fig.7. we can observe particles having 10x times larger size.

3. [Major] Lines 333-334 „The reason why WC phase is not detected may be that WC has a larger density than Ni60 so it is easy to sink to the bottom of the molten pool and there is no WC peak detected in the surface.” – what was the methodology of this phase analysis? On which part of the metallographic specimen it was conducted? W2C phase has a higher density than WC and it is detected in your investigation.

4. [Major] Lines 354-356 “Specimens 1, 4 and 7, prepared at identical scanning speeds but different powers, have relatively low hardness values, which differ greatly from those of other specimens, indicating that when the scanning speed is low, the cladding effect is unsatisfactory.”. What do you mean by “cladding effect is unsatisfactory”? There is no explanation the obvious differences in microhardness. Maybe the problems lays in the preparation of the samples? In Figure 7 we can see that participation of WC particles differs significantly in various samples, despite the fact the mass fraction was constant. For this reason, in my opinion, the technological aspects of the hard-faced samples (wear resistance) are very easy to undermine.

5. [Major] General remark: this paper is a sort of research report, it lacks wider discussion and references to the state of the art. Also, in some parts the discussion overstates the results, for example in 3.5. section, concerned with microstructural analysis, the authors write about differences in grain size, but in the presented photos the grainy structure is impossible to observe.

Overall: reject. 

Author Response

Thank you for your decision and constructive comments on my manuscript, and thank you very much for taking time out of your busy schedule to review the manuscript. After your suggestions, we have reorganized and rewritten the article, and added relevant literature, which is finally reflected in the abstract. I hope you can give me another chance. If you have any good suggestions, please let me know, and I will correct them in time.

Reviewer 2 Report

Comments and Suggestions for Authors

After going through the article titled "Effect of process parameters on microstructure and properties of laser cladding Ni60+30%WC coating on Q235 steel" I would like to put forward the following questions:

1. The reason for use of laser cladding should be clearly explained in the abstract itself.

2. What is the meaning of "The experiments carried out this time have some reference value for study on the application scope and useful life of Q235"? Please quantify important gains and compare improvement in percentage if any.

3. The abstract clearly lacks the novelty.

4. In the introduction, line 29, which industrial application? What is the advantage of using laser cladding? Can the surface be coated by some other method? How the laser cladding is better compared to some other known method of coating keeping in mind its cost. This should be answered to justify the use of laser cladding and its novelty. Will laser cladding be effectively used for curved surfaces since tests are being done on coupons.

5. The basis of choosing the parameters in Table 5 should be given namely the use of laser power, scanning speed and pass spacing.

6. What type of indenter was used to find out the microhardness? Vickers, Brinell, Berkovich??? This should be clearly stated. Also the number of tests conducted to ensure repeatability should be given.

7. What was the basis for choosing the tribo-test parameters in lines 134-137? Only the mass loss was measured? Can the authors report the wear rate or the wear coefficient?

8. The orthogonal array used was L9. The authors did not mention this. Also why was L9 array considered because the parameters chosen have interaction effects. This cannot be accomodated.

9. The text inside figure 3 cannot be read.

10. In the experimental details, it is not clearly mentioned as to what are the process parameters and what are the responses that are measured. This should be clearly stated.

11. I think the design of experiments is incorrect. Where is pass spacing in Table 6? The pass spacing is kept fixed. So how does it become a design variable? So it becomes a full factorial experiment where investigation has been done at all points.

12. Check the caption of Figure 9.

13. Lines 357 to 358, what is meant by large gaps in grain? Or is it higher grain size? Has the grain size been determined?

14. The lines 372-376 is very big and cannot be understood. Can it broken down to make it more understandable?

15. Why only mass loss of S5 is given? Where is the result for others? There are 9 specimens in all?

16. There is no worn surface analysis of the specimens and investigation of wear mechanisms. In such a case, lines 372-368 cannot be concluded.

17. In general, experimental designs are associated with optimization and statistical analyses. But I do not find such results.

18. Also the flow of the paper is quite confusing. What is the process parameter? What is the response? How have the authors found out better combination of parameters? Also some friction and wear tests are missing along with the wear mechanism.

19. In conclusion, the statement "There is good metallurgical bond between cladding layer and substrate" is not supported from experiments. How can this be concluded without performing scratch tests?

20. There is also no quantified gains presented in the conclusions.

Author Response

Thank you for your decision and constructive comments on my manuscript, and thank you very much for taking time out of your busy schedule to review the manuscript. After your suggestions, we have reorganized and rewritten the article, and added relevant literature, which is finally reflected in the abstract. 

Reviewer 3 Report

Comments and Suggestions for Authors

This paper focuses on the cladding process on structure property correlation of laser cladded Ni60+30%WC coating on Q235 steels. Though the work in this paper is of fundamental importance, it requires major revisions to reinforce the requirements for publication in Materials. Here are some major concerns regarding the claims and conclusions drawn from this study.

Author Response

(The authors gave the same response as above.)

Reviewer 4 Report

Comments and Suggestions for Authors

Comments

Title - Effect of process parameters on microstructure and properties 2 of laser cladding Ni60+30%WC coating on Q235 steel

This paper explores the enhancement of Q235 steel's durability by employing Ni60+30%WC composite powder via laser cladding technology. The study emphasizes the significant influence of scanning speed on cladding layer quality and provides valuable insights into the metallurgical bond between the cladding layer and the substrate, offering potential applications to extend the useful life of Q235 steel.

Here are my comments:

1.     The author should explicitly provide justification for the novelty of the research within the manuscript itself.

2.     You mentioned using various grit sizes for sanding, which could result in grit particles getting into the cladded surface. How did you remove them? Typically, it's advisable to perform an etching operation after sanding metal parts.

3.     In Figure 1, could you please clarify the location of Ni and WC?

4.     The laser scanning speed notably affects the surface quality of the product. How did you handle the laser speed by making energy density constant?

5.     Please specify the type of atmosphere maintained during the process, as moisture can significantly influence the working process.

Author Response

(The authors gave the same response as above.)

Reviewer 5 Report

Comments and Suggestions for Authors

The authors present a study related to the covering of surfaces through a method based on thermal treatment that uses the laser as the heat source. The purpose of this treatment is to improve the physical properties of the base material.

The topic addressed is in accordance with the topic of the journal. I have not identified similarities with other sources.

Overall, the work is well organized, easy to understand. The methods used are presented correctly. The conclusions have a support in the discussions related to the experimental data presented. The references are appropriate to the study and are up-to-date.

In my opinion, the work can be published after a slight adjustment, namely:

In the first paragraph from the Conclusions we have the letter r missing from the word laser.

In figure 4, the font used for the information associated with the axes is too small.

Also, all figures should be scaled in such way to be uniform regarding the front aspect, if it's possible. For example, between Figs 7 and 8 labels of the samples are using different font sizes, so the aestethical aspect of images is perturbed. In figures 9-11 fonts used for the axis are bigger than the font used in text, again a disturbed aesthetic view are resulted.

Author Response

(The authors gave the same response as above.)

Round 2

Reviewer 1 Report

Comments and Suggestions for Authors

The authors did the corrections. I recommend the publication.

Reviewer 2 Report

Comments and Suggestions for Authors

The authors have addressed the comments.